# Bilateral Deep Vein Thrombosis and Pulmonary Embolism Due to Right Common Iliac Artery Aneurysm with a Contained Rupture

**DOI:** 10.3390/medicina58030421

**Published:** 2022-03-13

**Authors:** Foteini Malli, Ilias E. Dimeas, Sotirios I. Sinis, Eleni Karetsi, Petroula Nana, George Kouvelos, Konstantinos I. Gourgoulianis

**Affiliations:** 1Respiratory Medicine Department, School of Medicine, University of Thessaly, Biopolis (Mezourlo), 41110 Larissa, Greece; dimel13@hotmail.com (I.E.D.); sinis1995@windowslive.com (S.I.S.); ekaretsi@gmail.com (E.K.); kgourg@uth.gr (K.I.G.); 2Respiratory Disorders Lab., Faculty of Nursing, University of Thessaly, 41500 Larissa, Greece; 3Vascular Surgery Department, Larissa University Hospital, School of Medicine, University of Thessaly, 41110 Larissa, Greece; petr.nana7@hotmail.com (P.N.); geokouv@gmail.com (G.K.)

**Keywords:** venous thromboembolism, pulmonary embolism, deep venous thrombosis, iliac artery aneurysms, endovascular repair

## Abstract

Venous thromboembolism (comprising deep venous thrombosis and/or pulmonary embolism) is a common disease, often of multifactorial cause. Focal iliac artery aneurysms are relatively rare, and only a few reports exist in the literature describing patients with venous thromboembolism resulting from venous floe disruption due to iliac artery aneurysm. Thus, we report a case of a 65-year-old male presenting with pulmonary embolism and bilateral deep vein thrombosis associated with a contained rupture of the right common iliac artery aneurysm.

## 1. Introduction

Venous thromboembolism (VTE) is a relatively common disease that accounts as the third leading cause of cardiovascular mortality with an increasing incidence in recent years [1]. The predisposing factors of VTE may be acquired and/or hereditary, while VTE events are often of combined multifactorial mechanisms [2]. History of a prior VTE and cancer are common risk factors that are present in 26% and 22.3% (respectively) of patients [3]. The prevalence of PE increases progressively with the number of risk factors present; subjects with ≥4 predisposing factors experience PE in 30.43% [4]. Early diagnosis of an acute event is related to decreased mortality and, thus, the identification of patients at risk is mandatory. Herein, an unusual case of bilateral deep vein thrombosis (DVT) and pulmonary embolism (PE) in the setting of a ruptured right common iliac artery aneurysm (CIA) due to common iliac vein compression is presented.

## 2. Case Presentation

A 65-year-old man presented to the emergency department with complaints about a self-limited transient loss of consciousness (syncope). The patient reported that two days ago, he developed pain and oedema in his left limb with concomitant dyspepsia associated with postprandial abdominal pain. Additionally, he complained about colic pain located to the pelvis that initiated a month ago. The pain irradiated to the perineum. His medical history was null and reported no trauma. During the initial evaluation, the patients’ vital signs were normal. The laboratory tests revealed a normal cell blood count and an elevated D-dimers’ value (6506 ng/mL, upper normal limit 255 ng/mL). He was subjected to duplex ultrasonography that revealed echogenic material, loss of compressibility and continuous Doppler signal, consistent with DVT extending from the popliteal to the left common iliac vein. The patient was subjected to computed tomography (CT) pulmonary angiography that depicted (Figure 1) bilateral PE involving the main pulmonary arteries. Transthoracic echocardiography did not show any signs of right heart strain. The patient received low molecular weight heparin and was admitted to the Respiratory Medicine Department of the University Hospital of Larissa. During the second day of hospitalization, he complained about worsening pelvic pain and was subjected to CT of the abdomen; the CT revealed a 5.3 cm aneurysm of the right CIA with crescents of high-attenuation and surrounding soft-tissue density and fluid accumulation consisting of a contained rupture. Furthermore, a dilatation of the infra-renal aorta was revealed (3.0 cm). The aneurysm extended from the aortic bifurcation down to the right iliac bifurcation. Additionally, the right common iliac vein and inferior vena cava were thrombosed. Due to these findings, prompt vascular surgeon consultation was obtained. The patient was subjected to endovascular repair, using percutaneous access, coiling of the right internal iliac artery and aneurysm exclusion using a bifurcated low-profile endograft (AFX-2, Endologix LLC, 2 Musick, Irvine, CA 92618, USA) and a right limb extension down to the right external iliac artery (Endurant, 710 Medtronic Parkway, Minneapolis, MN 55432-5604, USA). The intra-operative completion angiography confirmed aneurysm exclusion with no endoleak. The procedure was uneventful, and the patient was started on rivaroxaban in the 2nd post-operative day. He was discharged on the 3rd post-operative day under rivaroxaban 15 mg twice a day. He is currently under follow-up (9 months post the acute event) by both pulmonologists and vascular surgeons. He is currently receiving rivaroxaban 10 mg once daily with no signs of recurrence or bleeding. Villalta score was 2, suggesting the absence of post-thrombotic syndrome [5]. Screening for occult predisposing factors (i.e., cancer, antiphospholid syndrome, hereditary thrombophilia) was negative.

## 3. Discussion

Focal CIA aneurysms comprise 0.4–1.9% of all aneurysms with an incidence of 0.03% in the general population [6,7]. CIA aneurysms are more commonly found in males with a 7:1 ratio to women and are usually diagnosed in the 7th to 8th decade of life. Most CIA aneurysms are diagnosed incidentally (i.e., the patient is asymptomatic), while symptoms may arise due to rupture or from the compression or erosion of adjacent structures such as the sacral plexus, ureter or iliac vein. Common presentations include lower abdominal pain, pyelonephritis, pain on defecation, and paraesthesia of the lower extremities. In the present case, the clinical symptoms were quite atypical; the patient reported pelvic and post-prandial pain. The presence of concomitant DVT could affect the diagnostic approach and cover the underlying ruptured aneurysm.

Only a few reports exist in the literature on CIA aneurysm presenting with DVT and/or PE. Most of these cases are right sided and are associated with right DVT [8], while most published reports present unruptured aneurysms. The occurrence of DVT in patients with CIA aneurysms results mainly due to chronic compression of the iliac veins between the overlying CIA and the adjacent structures. The condition is rare, and few cases have been previously reported [9,10,11]. The pressure phenomenon usually results in ipsilateral or contralateral DVT, which are probably due to chronic venous stasis and/or endothelial damage because of the repeatable pulsatile force of the CIA aneurysm as an analogue to the May Thurner syndrome [12]. In the present case, a contained rupture of a CIA aneurysm affected the right and left iliac vein flow and provoked an associated PE.

The management of unruptured CIA aneurysms aims to exclude aneurysmal flow and prevent rupture and involves both endovascular and open repair techniques [13]. The choice of treatment is a matter of ongoing controversy with some researchers, suggesting that open repair is preferred over endovascular management [14]. However, in patients with coexisting DVT and PE, open surgical repair may be challenging due to the higher rates of postoperative complications (i.e., cardiac events). In the present case, which involved an associated acute iliac DVT, any surgical manoeuvre during vessel dissection could result in a new embolization and a massive, probably lethal, PE.

## 4. Conclusions

In conclusion, iliac artery aneurysms may represent an unusual cause of VTE. Patients with VTE and abdominal symptoms may be subjected to further investigations, as the possibility of an underlying arterial aneurysm cannot be excluded.

## Figures and Tables

**Figure 1 medicina-58-00421-f001:**
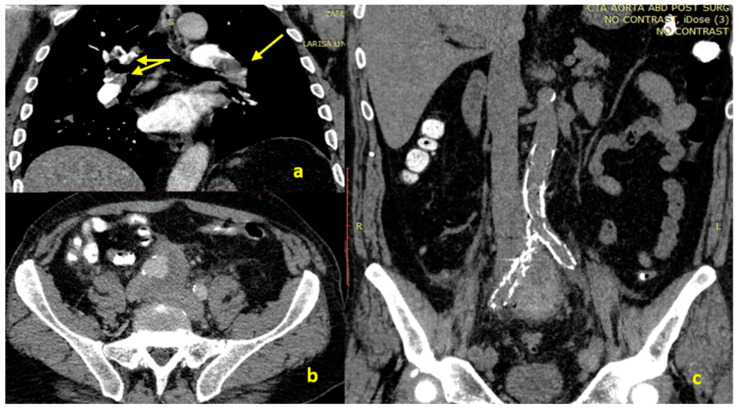
CT pulmonary angiography showing pulmonary embolism (**a**), coronal 63 view of CT of the abdomen presenting the contained rupture of the aneurysm (**b**), CT of 64 the abdomen after stent placement (**c**).

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
