# Peer review of "Bilateral Deep Vein Thrombosis and Pulmonary Embolism Due to Right Common Iliac Artery Aneurysm with a Contained Rupture"

_medicina, 2022, doi:10.3390/medicina58030421_

Round 1

Reviewer 1 Report

Thank you for this interesting case report

I would like to suggest the followings: 

  1. clinical background or medical comorbidities of the patient should be mentioned
  2. any specific cancer screening like colonoscopy, if done can be mentioned 
  3. if any arrow can indicate the thrombus in the CT scan will be helpful

Author Response

We are very grateful for the instructive and helpful comments. Our responses to their comments are as follows. We are happy to assist with any further clarifications if needed.

Point 1. clinical background or medical comorbidities of the patient should be mentioned
Reponse 1. The patient had no comorbidities or previous hospitalizations. We have indicated this in the text “His medical history was null and reported no trauma”.
Point 2. any specific cancer screening like colonoscopy, if done can be mentioned 
Response 2. The patient did not present any signs indicative of cancer (i.e. systemic symptoms), his CBC was within normal limits 1 month after his surgery, his Sedimentation Rate was normal and his PSA levels were within normal range. There were no signs indicative of cancer in his CT scan of the lung or the abdomen. At his follow-up there are still no signs of occult cancer. The aforementioned data are briefly described in the Results section as follows: “Screening for occult predisposing factors (i.e. cancer, antiphospholid syndrome, hereditary thrombophilia) was negative”.  
Point 3. if any arrow can indicate the thrombus in the CT scan will be helpful
Response 3. Following the reviewers’ concerns, we have added arrows to indicate the thrombus in the CT scan.   

Reviewer 2 Report

Dear Authors:

This is a case report. About aneurysm therapy and pulmonary embolism relation. Good finding.

   Thank you.

Author Response

We are very grateful for the instructive and helpful comments. Our responses to their comments are as follows. We are happy to assist with any further clarifications if needed.

Point 1. Introduction: Need more words.
Response 1. We have added data in the Introduction section of the revised manuscript.

Point 2. Case presentation:Need language edited.
Response 2. We have rephrased the text in the revised manuscript. 

Point 3. Find 5.3 cm the right aneurysm, subjected to endovascular repair. Receiving rivaroxaban dose from 15mg to 10mg after surgery. No recurrence, No bleeding. Could you tell me this surgery operation whether like percutaneous transluminal coronary angioplasty, PTCA? The right panel cited (C). And Legend.
Response 3. The patient was subjected to endovascular repair through the femoral artery. 

Point 4. Discussion: Need language edited. This case report aims to exclude aneurysmal flow and prevent rupture and involves both endovascular and open repair techniques. Could you show me about aneurysmal flow and pulmonary embolism relationship? After surgery.
Response 4. We would like to thank the reviewer for the instructive comments. We have edited the text in the revised manuscript. Aneurysmal flow could be one element of Virchow's triad but it hasn't be examined thourghly and this is the reason we haven't really mentioned it in the manuscript.

Point 5. References: Change 1, 2, 4, 5, 8, 9, 10
Response 5. We have edited the refernces appropriately.

Reviewer 3 Report

Nice paper on a case report of concomitant rupture CIA aneurysm and bilateral DVT, a rare occurence. 

The paper il well written and all diagnosis and therapeutical elements are clearly enounced. 

page 3, line87: aims instead of aim

Author Response

We would like to thank the reviewer for the kind comments. Our response to his/her comment is as follows. We are happy to assist with any further clarifications if needed.

Point 1. page 3, line87: aims instead of aim.
Response 1. We would like to thank the reviewer for the helpful suggestion. We have changed the text accordingly.